

# Satellite-derived Ecosystem Functional Types capture ecosystem functional heterogeneity at regional scale

Beatriz P. Cazorla[1,2*], Ana Meijide[3], Javier Cabello[1,4], Julio Peñas[1,5], Rodrigo Vargas[6], Javier Martínez-López[1,2,7], Leonardo Montagnani[8], Alexander Knohl[9], Lukas Siebicke[9], Benimiano Gioli[10], Jiří Dušek[11], Ladislav Šigut[11], Andreas Ibrom[12], Georg Wohlfahrt[13], Eugénie Paul-Limoges[14], Kathrin Fuchs[15], Antonio Manco[16], Marian Pavelka[11], Lutz Merbold[17], Lukas Hörtnagl[18], Pierpaolo Duce[10], Ignacio Goded[19], Kim Pilegaard[12], Domingo Alcaraz-Segura[1,2,4]

[1]Andalusian Center for Global Change – Hermelindo Castro (ENGLOBA), University of Almería, 04120 Almería, Spain.
[2]Interuniversity Institute of Earth System Research in Andalusia (IISTA), 18006 Granada, Spain.

[3]Environment Modeling, Institute of Crop Science and Resource Conservation, University of Bonn, Niebuhrstraße 1a, 53113 Bonn, Germany.
[4]Department of Biology and Geology, University of Almería, 04120 Almería, Spain.
[5]Department of Botany, University of Granada, 18006 Granada, Spain.
[6]Department of Plant and Soil Sciences, University of Delaware, Newark, Delaware, USA.
[7]Department of Ecology, University of Granada, 18006 Granada, Spain.
[8]Faculty of Agricultural, Environmental and Food Sciences, Free University of Bolzano, Italy.
[9]University of Goettingen, Bioclimatology, Büsgenweg 2, 37077 Göttingen, Germany.
[10]CNR – IBE, Institute of Bioeconomy, Firenze, Italy.
[11]Department of Matters and Energy Fluxes, Global Change Research Institute CAS, Brno, Czech Republic.
[12]Department of Resource and Environmental Engineering, Technical University of Denmark, Kongens Lyngby, Denmark.
[13]Department of Ecology, University of Innsbruck, Innsbruck, Austria.
[14]Department of Geography, University of Zurich, Switzerland.
[15]Karlsruhe Institute of Technology, Institute of Meteorology and Climate Research - Atmospheric Environmental Research, Garmisch-Partenkirchen, Germany.
[16]Institute for Agriculture and Forestry Systems in the Mediterranean (ISAFoM), National Research Council of Italy, P.le E. Fermi 1-Loc, Porto del Granatello 1, 80055 Portici, Italy.
[17]Department of Agroecology and Environment, Reckenholzstrasse 191, 8046 Zurich, Switzerland.
[18]Department of Environmental System Science, Institute of Agricultural Science, Zurich, Switzerland.
[19]European Commission, Joint Research Centre, Ispra, Italy.

*Correspondence to*: Beatriz P. Cazorla (b.cazorla@ual.es)

**Abstract.** Assessing ecosystem functioning is crucial for managing and conserving ecosystems and their services. Numerous ways to evaluate ecosystem functioning have been developed, using species traits, such as Plant Functional Types (PFTs), flux measurements with Eddy Covariance (EC) technique, and remote sensing techniques. We propose that the spatial heterogeneity in ecosystem functioning at a regional scale can be assessed and monitored using satellite-derived Ecosystem Functional Types (EFTs): groups of ecosystems or patches of the land surface that share similar dynamics of matter and energy exchanges. We hypothesize that, as observed for PFTs, different EFTs should have distinct patterns and magnitudes of Net Ecosystem Exchange (NEE) of carbon dioxide measured using the EC technique. We derived EFTs based on the 2001-2014 time-series of satellite images of the Enhanced Vegetation Index (EVI) and compared them with NEE measurements (derived from in situ



field observations using the EC technique) across 50 European sites. Our results show that distinct EFTs classes display significantly different dynamics and magnitudes of NEE and that EFTs perform marginally better than PFTs in explaining NEE regional patterns. Land-cover maps based on PFTs are difficult to update on an annual basis and are not sensitive to changes in ecosystem performance (e.g., droughts or pests) that do involve short-term changes in PFT composition. In contrast, satellite-derived EFTs are sensitive to short-term changes in ecosystem performance. Satellite-derived EFTs are an ecosystem

functional classification built from satellite observations that allow the identification of homogeneous land patches in terms of ecosystem functions, e.g., ecosystem net productivity measured on the ground as NEE. Satellite-derived EFTs can be recalculated annually, providing a straightforward way to assess and monitor interannual changes in ecosystem functioning and functional diversity.

# 1 Introduction

Ecosystem functioning and functional diversity are critical issues of current ecological research (Jax, 2010; Violle et al. 2014, 2017; Tilman et al. 2014; Pettorelli et al. 2018; Villarreal et al. 2018; Malaterre et al. 2019; Díaz et al. 2020). Quantifying, monitoring, and understanding ecosystem functioning help provide insights into the management and conservation of ecosystems and their services (Cabello et al. 2012; Pettorelli et al. 2018; Nicholson et al. 2021). Variables capable of describing ecosystem functioning at regional to global scales are needed to define essential biodiversity variables to monitor biodiversity

status (Pereira et al. 2013; Jetz et al. 2019), to advance in the definition of critical but still unassessed planetary boundary (Steffen et al. 2015; Richardson et al. 2023), and to quantify their associated ecosystem services (Costanza et al. 1997; Balvanera et al. 2017).

There are multiple ways to evaluate ecosystem functioning, from concepts such as species traits or Plant Functional Types (PFTs) to direct observation techniques such as eddy covariance (EC) and remote sensing. Traditionally, studies on ecosystem

functioning were approached by grouping species into PFTs based on structural (e.g., biotypes), phylogenetic (e.g., coniferous), or functional species traits (e.g., metabolic pathway) that were linked to biological processes (Lavorel et al. 2002, 2007). For instance, the PFT approach has been widely used in land-cover mapping and dynamic vegetation models to simplify the continuum of species traits into a reduced number of discrete categories suitable for regional-to-global synthesis and modeling studies (Wullschleger et al. 2014). However, this simplification can lead to information loss (Funk et al. 2017) and

may not be capable of predicting the overall ecosystem functioning (Virtanen, 2017; Thomas et al. 2019). Another more recent way to evaluate ecosystem functioning is by using EC (Reichstein et al. 2014; Migliavacca et al. 2021). This method uses high-frequency wind and scalar mixing ratio data for calculating the Net Ecosystem $CO_2$ Exchange (NEE) between the land surface and the atmosphere at the field scale (Baldocchi et al. 2001, 2020). This approach is widely used and regional (e.g., AmeriFlux, AsiaFlux, ICOS, NEON), and a global network of EC measurements has been formed (e.g., FLUXNET) (Franz

et al. 2018; Knox et al. 2019). Although FLUXNET has provided unprecedented information on the carbon, water, and energy exchange between the earth's surface and the atmosphere, these measurements still show limitations to assessing ecosystem



functioning at regional or global scales due to their small footprints (essentially considered as point-scale data (Chu et al. 2021) and a lack of representativity (Villarreal et al. 2018, 2021). In parallel, advances in remote sensing are providing new opportunities to quantify ecosystem functioning and functional diversity from regional to global scales (Rocchini et al. 2018;

Skidmore et al. 2021). Consequently, combining field-based measurements (e.g., EC) with remote sensing data may allow for better information integration across multiple spatial and temporal scales (Running et al. 1999; Wang et al. 2017). Indeed, multiple studies have aimed to derive global maps combining flux measurements with earth observation data, although challenges and limitations still need to be addressed (e.g., FLUXCOM; Huang et al. 2019; Jung et al. 2020; Liu et al. 2023; Pacheco-Labrador et al. 2023; Gomarasca et al. 2024; Nelson et al. 2024).

Ecosystem functioning and functional diversity at the regional scale can be assessed using satellite-derived Ecosystem Functional Types (EFTs) (Paruelo et al. 2001). Conceptually, EFTs are defined as patches of the land surface that share similar dynamics of matter and energy exchanges between the biota and the physical environment (Alcaraz-Segura et al. 2006, 2013; Cazorla et al. 2020, 2021, 2023). The concept of EFT is equivalent to the concept of PFTs but applied to a higher level of biological organization. That is, just like plant species can be grouped based on shared functional traits (e.g., growth rates,

nitrogen fixation) into PFTs, ecosystems can be grouped based on their common functional dynamics (e.g., productivity, seasonality, phenology) into EFTs (Paruelo et al. 2001). Remote sensing has been empirically applied to identify EFTs, mainly through spectral indices related to carbon dynamics (Paruelo et al., 2001; Alcaraz-Segura et al., 2006; Ivits et al., 2013), but also incorporating other functional attributes such as evapotranspiration, surface temperature, and albedo (e.g., Fernández et al. 2010; Pérez-Hoyos et al. 2014) or soil characteristics based on their greenhouse gas flux dynamics (Petrakis et al. 2018).

Other authors have used EFTs to: describe large-scale functional biogeographical patterns (Ivits et al. 2013; Cazorla et al. 2021), assess the representativeness of environmental observatory networks (Villarreal et al. 2018, 2019), assess the ecosystem functional diversity (Alcaraz-Segura et al. 2013; Liu et al. 2023; Amstrong et al. 2024), evaluate the effects of land-use changes on ecosystem functioning (Oki et al. 2012; Domingo-Marimon et al. 2024), improve weather forecasting (Lee et al. 2013; Müller et al. 2014) and species distribution/abundance models (Arenas-Castro et al. 2018, 2019), and to identify geographic

priorities for biodiversity conservation (Cazorla et al. 2020).

So far, EFTs have been identified from satellite remote sensing data. However, whether such top-down-identified EFT classes are biologically meaningful in ecological processes measured on the ground, such as biogeochemical fluxes, remains untested. That is, whether satellite-derived EFT classes differ in their exchanges of energy and matter between ecosystems and the atmosphere. Therefore, linking satellite-derived EFTs identified at large scales to biogeochemical fluxes measured at the site

level could help strengthen the ecological significance of the EFT patterns for ecosystem modeling and functional diversity assessments remotely, as it provides empirical evidence for using the concept at these scales.

This study aims to provide field-based empirical evidence for using satellite-derived EFTs as descriptors of regional heterogeneity in ecosystem functioning measured on the ground (i.e., seasonal dynamics of NEE). We hypothesize that satellite-derived EFTs classes significantly differ in their exchanges of energy and matter with the atmosphere from each other,

in the same way as estimated with in situ field observations. Here, we propose that different satellite-derived EFTs classes



display significantly different NEE measured using the EC technique, while sites under the same EFT should exhibit similar NEE dynamics. To achieve our goal, we used publicly available data across continental Europe, given its high density of EC sites, 1) to characterize the regional patterns of ecosystem functioning using satellite-derived EFTs; 2) to assess whether different satellite-derived EFTs correspond to different NEE dynamics measured on the ground with the EC technique; and 3) to assess how EFTs perform compared to traditional PFTs to discriminate different NEE dynamics.

## 2 Material and methods

### 2.1 Study area

We used NEE information from continental Europe as it has one of the largest densities of EC sites worldwide (Table 1). The sites were distributed across four biogeographical regions (EEA 2016): Mediterranean (12 sites), Continental (21 sites), Atlantic (9 sites), and Alpine (8 sites). Only sites with a long-term (i.e., from 3 to 14 years) NEE time-series were included in the analysis (detailed below).


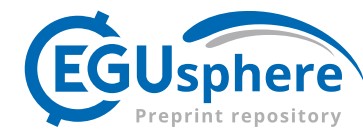

**Table 1.** Main characteristics of the 50 Eddy Covariance (EC) sites in the study area. Data from FLUXNET 2015 dataset

| ID | Site | Country | PFT | EFT code | Biogeographical region | n years (2001-2014) | Elevation (m) | Latitude | Longitude |
|---|---|---|---|---|---|---|---|---|---|
| AT-Neu | Neustift/Stubai Valley | Austria | Grasslands | Da2 | Alpine | 11 (2002-2013) | 970 | 47.116 | 11.317 |
| BE-Bra | Brasschaat (De Inslag Trees) | Belgium | Mixed Trees | Cc1 | Atlantic | 14 (2001-2014) | 16 | 51.309 | 4.520 |
| BE-Lon | Lonzee | Belgium | Croplands | Ba1 | Atlantic | 11 (2004-2014) | 167 | 50.552 | 4.744 |
| BE-Vie | Vielsalm | Belgium | Mixed Trees | Bc1 | Continental | 14 (2001-2014) | 439 | 50.305 | 5.998 |
| CH-Cha | Chamau grassland | Switzerland | Grasslands | Db1 | Continental | 10 (2005-2014) | 393 | 47.210 | 8.410 |
| CH-Dav | Davos-Seehorn forest | Switzerland | Evergreen Needleleaf Trees | Ac2 | Alpine | 14 (2001-2014) | 1639 | 46.815 | 9.855 |
| CH-Fru | Fruebuel grassland | Switzerland | Grasslands | Da2 | Continental | 10 (2005-2014) | 982 | 47.115 | 8.537 |
| CH-Lae | Laegeren | Switzerland | Mixed Trees | Da1 | Continental | 11 (2004-2014) | 689 | 47.478 | 8.365 |
| CH-Oe1 | Oensingen1 grass | Switzerland | Croplands | Cb1 | Continental | 7 (2002-2008) | 450 | 47.285 | 7.731 |
| CZ-BK1 | BilyKriz-Beskidy Mountains | Czech Republic | Evergreen Needleleaf Trees | Cc1 | Continental | 11 (2004-2014) | 875 | 49.502 | 18.536 |
| CZ-BK2 | BilyKriz-grassland | Czech Republic | Mixed Trees | Ac1 | Alpine | 9 (2004-2012) | 855 | 49.494 | 18.542 |





| ID | Site | Country | PFT | EFT code | Biogeographical region | n years (2001-2014) | Elevation (m) | Latitude | Longitude |
|---|---|---|---|---|---|---|---|---|---|
| CZ-wet | CZECHWET | Czech Republic | Wetlands | Ba1 | Continental | 9 (2004-2012) | 426 | 49.024 | 14.770 |
| DE-Akm | Anklam | Germany | Wetlands | Ba1 | Continental | 5 (2010-2014) | -1 | 53.866 | 13.683 |
| DE-Geb | Gebesee | Germany | Croplands | Ba1 | Continental | 14 (2001-2014) | 161 | 51.100 | 10.914 |
| DE-Gri | Grillenburg-grass station | Germany | Grassland | Da2 | Continental | 11 (2004-2014) | 385 | 50.949 | 13.512 |
| DE-Hai | Hainich | Germany | Mixed Trees | Ca1 | Continental | 12 (2001-2012) | 430 | 51.079 | 10.452 |
| DE-Kli | Klingenberg | Germany | Croplands | Ba1 | Continental | 11 (2004-2014) | 478 | 50.892 | 13.522 |
| DE-Lkb | Lackenberg | Germany | Evergreen Needleleaf Trees | Ab2 | Continental | 5 (2009-2013) | 1308 | 49.099 | 13.304 |
| DE-Lnf | Leinefelde | Germany | Deciduous Broadleaf Trees | Da1 | Continental | 11 (2002-2012) | 451 | 51.328 | 10.367 |
| DE-RuR | Rollesbroich | Germany | Grasslands | Da2 | Continental | 4 (2011-2014) | 515 | 50.621 | 6.304 |
| DE-RuS | Selhausen Juelich | Germany | Croplands | Cb1 | Atlantic | 4 (2011-2014) | 103 | 50.865 | 6.447 |
| DE-Seh | Selhausen | Germany | Croplands | Cb1 | Atlantic | 4 (2007-2010) | 103 | 50.870 | 6.449 |
| DE-Spw | Spreewald | Germany | Mixed Trees | Ca1 | Continental | 5 (2010-2014) | 61 | 51.892 | 14.033 |
| DE-Tha | Tharandt-Anchor Station | Germany | Evergreen Needleleaf Trees | Bc1 | Continental | 14 (2001-2014) | 385 | 50.963 | 13.566 |
| DK-Eng | Enghave | Denmark | Croplands | Ca1 | Continental | 4 (2005-2008) | 10 | 55.690 | 12.191 |



| ID | Site | Country | PFT | EFT code | Biogeographical region | n years (2001-2014) | Elevation (m) | Latitude | Longitude |
|---|---|---|---|---|---|---|---|---|---|
| DK-Sor | Soroe-LilleBogeskov | Denmark | Deciduous Broadleaf Trees | Da1 | Continental | 14 (2001-2014) | 40 | 55.485 | 11.644 |
| ES-Amo | Amoladeras | Spain | Shrublands | Ad4 | Mediterranea | 6 (2007-2012) | 58 | 36.833 | -2.252 |
| ES-LJu | Llano de los Juanes | Spain | Shrublands | Ad1 | Mediterranea | 10 (2004-2013) | 1600 | 36.926 | -2.752 |
| FR-Fon | Fontainebleau | France | Deciduous Broadleaf Trees | Da1 | Atlantic | 10 (2005-2014) | 103 | 48.476 | 2.780 |
| FR-Gri | Grignon | France | Croplands | Cc1 | Atlantic | 11 (2004-2014) | 125 | 48.844 | 1.951 |
| FR-Pue | Puechabon | France | Mixed Trees | Cd1 | Mediterranea | 14 (2001-2014) | 270 | 43.741 | 3.595 |
| IT-BCi | Borgo Cioffi | Italy | Croplands | Db4 | Mediterranea | 11 (2004-2014) | 20 | 40.523 | 14.957 |
| IT-CA1 | Castel d'Asso1 | Italy | Croplands | Bd1 | Mediterranea | 4 (2011-2014) | 200 | 42.380 | 12.026 |
| IT-CA2 | Castel d'Asso2 | Italy | Croplands | Cb1 | Mediterranea | 4 (2011-2014) | 200 | 42.377 | 12.026 |
| IT-CA3 | Castel d'Asso 3 | Italy | Croplands | Bd1 | Mediterranea | 4 (2011-2014) | 197 | 42.380 | 12.022 |
| IT-Col | Collelongo-Selva Piana | Italy | Deciduous Broadleaf Trees | Da1 | Alpine | 14 (2001-2014) | 1560 | 41.849 | 13.588 |
| IT-Cpz | Castelporziano | Italy | Evergreen Needleleaf Trees | Dd1 | Mediterranea | 9 (2001-2009) | 68 | 41.705 | 12.376 |
| IT-Lav | Lavarone (after 3/2002) | Italy | Evergreen Needleleaf Trees | Bc1 | Alpine | 12 (2003-2014) | 1353 | 45.956 | 11.281 |
| IT-MBo | Monte | Italy | Grasslands | Aa1 | Alpine | 11 (2003-2013) | 1550 | 46.014 | 11.045 |





Bondone

| ID | Site | Country | PFT | EFT code | Biogeographical region | n years (2001-2014) | Elevation (m) | Latitude | Longitude |
|---|---|---|---|---|---|---|---|---|---|
| IT-Noe | Sardinia/Arca di Noe | Italy | Shrublands | Ad1 | Mediterranea | 11 (2004-2014) | 25 | 40.606 | 8.151 |
| IT-Ro1 | Roccarespampani1 | Italy | Deciduous Broadleaf Trees | Da1 | Mediterranea | 8 (2001-2008) | 235 | 42.408 | 11.930 |
| IT-Ro2 | Roccarespampani2 | Italy | Deciduous Broadleaf Trees | Da1 | Mediterranea | 11 (2002-2012) | 160 | 42.390 | 11.920 |
| IT-SRo | San Rossore | Italy | Evergreen Needleleaf Trees | Cd3 | Mediterranea | 12 (2001-2012) | 6 | 43.727 | 10.284 |
| IT-Tor | Torgnon | Italy | Grassland | Aa1 | Alpine | 7 (2008-2014) | 1260 | 45.844 | 7.578 |
| NL-Hor | Horstermeer | Netherlands | Mixed Trees | Da1 | Atlantic | 8 (2004-2011) | 2 | 52.240 | 5.071 |
| NL-Loo | Loobos | Netherlands | Evergreen Needleleaf Trees | Bd2 | Atlantic | 14 (2001-2014) | 25 | 52.166 | 5.743 |



## 2.2 Satellite-derived Ecosystem Functional Types (EFTs)

To characterize the regional heterogeneity in ecosystem functioning across continental Europe, we identified EFTs based on the 2001-2014 time-series of satellite images of the Enhanced Vegetation Index (EVI) captured by the MODIS-Terra sensor.

These images (MOD13Q1.C006 product) provide a maximum composite EVI value every 16 days at a ~230 m spatial resolution. EVI is a proxy for canopy greenness, vegetation carbon gains, or primary production (Huete et al. 1999). Based on the approach by Alcaraz-Segura et al. (2013), we identified EFTs using three biologically meaningful metrics of the EVI seasonal dynamics: the EVI annual mean (EVI_mean; an estimator of annual primary production), the EVI seasonal standard deviation (EVI_SD; a descriptor of seasonality), and the date of maximum EVI (EVI_DMAX; an indicator of phenology). We

chose to use MODIS data instead of other satellites with higher spatial resolution (e.g., Landsat or Sentinel-2) because MODIS has several advantages in terms of data availability and quality (e.g. more years of data and cloud-free image every 16-days) along the time series (see S1).

The range of values of each EVI metric was divided into four intervals, giving a potential number of 64 EFTs ($4 \times 4 \times 4$). For EVI_DMAX, the four intervals agreed with the four seasons of the year. For EVI_mean and EVI_SD, we extracted the first,

second, and third quartiles for each year and then calculated their interannual average for the 14 years. To name EFTs, we used two letters and a number: the first capital letter indicates net primary production (EVI_mean), increasing from A to D; the second small letter represents seasonality (EVI_SD), decreasing from a to d; the numbers are a phenological indicator of the growing season (EVI_DMAX), with values 1-spring, 2-summer, 3-autumn, 4-winter. To summarize the ecosystem functional diversity of the 2001–2014 period, we calculated the dominant EFT (i.e., the mode value for each pixel) of these years.

## 2.3 Eddy covariance (EC) sites for net ecosystem exchange (NEE)

To obtain NEE fluxes, 50 EC sites were selected across our study area from the FLUXNET2015 dataset (Table 1). The FLUXNET network (Baldocchi et al. 2001, 2020) provides high-quality, community-based, global data on $CO_2$, $H_2O$, and energy exchanges between the biosphere and the atmosphere measured using the EC technique (Baldocchi, 2003). We used data of NEE of $CO_2$ (NEE_VUT_REF, gC m-2 d-1) from the FLUXNET2015 database. We selected data from

FLUXNET2015 because they are publicly available and offer benefits in terms of standardized methodology. FLUXNET2015 incorporates NEE measurements along with a quality flag based on an annually determined Variable Ustar Threshold (VUT), which is selected to maximize model efficiency (MEF) (Pastorello et al. 2020). The MEF analysis is repeated for each one of the half-hourly data (Baldocchi et al. 2001, 2020). We selected sites that: (a) were located in our study area; (b) provided more than three consecutive years of data over the 2001-2014 period; (c) provided daily averages of NEE calculated from half-

hourly data; and (d) had quality control information (i.e., NEE_VUT_REF data with quality control flag QC > 1 were removed since they represent medium and poor quality gap-filled data).



We applied Discriminant Analysis (DA) to assess whether different satellite-derived EFT classes correspond to different NEE dynamics and whether sites under the same EFT exhibit similar NEE dynamics (S2). The DA allowed us to examine the homogeneity within each EFT class and the differences among EFT classes based on the annual dynamics of NEE as a predictor variable (Williams,1981, 1983). We selected the EFT where each EC site was located and its corresponding interannual average of the seasonal cycle of NEE for the available years. EC sites fluxes were regarded as the ground truth standard against which the satellite data were compared to calculate five performance metrics: Kappa, Accuracy, Precision, Recall, and F1 score (Table 2).

**Table 2.** Metrics, interpretations, and equations used to evaluate and compare results from the discriminant analysis, Pr(a) is the relative observed agreement between observations, and Pr(e) is the hypothetical probability of agreement by chance. True Positives are correctly classified as positive, True Negative are correctly classified as negative, Positives are all positives including false positives (i.e., including falsely classified as positive, Type I error) and, Negatives are all negatives including false negatives (i.e. falsely classified as negative, Type II error). All performance metrics oscillate between 0 (disagreement) and 1 (maximum agreement).

| Metric | Meaning | Equation |
|---|---|---|
| **Kappa** | Measures the percentage of data values in the main diagonal of the contingency table and adjusts these values for agreement that could be expected due to chance alone | $K = Pr(a)-Pr(e) / 1-Pr(e)$ |
| **Accuracy** | Degree of closeness of measurements of a quantity to that quantity's true value | Accuracy = (True Positives + True Negatives )/ (Positives+Negatives) |
| **Precision** | Fraction of relevant instances among the retrieved instances (also called positive predictive value, i.e., how many EFTs were well discriminated) | Precision = True Positives / (True Positives+False Positives) |
| **Recall** | Fraction of relevant instances that have been retrieved over the total amount of relevant instances | Recall = True Positives / (True Positives+False Negatives) |



| F1 | Considers both the Precision and the Recall of the test to compute the score | F1 score= 2 × (Precision × Recall) / (Precision + Recall) |
|---|---|---|

## 2.4 Comparing how EFTs and PFTs discriminate different NEE dynamics

The PFT corresponding to each EC site was assigned by each of their principal investigators using the International Geosphere-
Biosphere Programme (IGBP, 1992). Subsequently, we verified the assigned PFTs using the MODIS MCD12Q1 land cover
product. The PFT categories present in the EC sites were: cropland (15 sites), deciduous broadleaf trees (6), evergreen
needleleaf trees (10), grassland (5), mixed trees (8), shrubland (3), and wetland (1) (Table 1).

During the comparison of the performance of PFTs and EFTs to discriminate the seasonal dynamics of NEE, we considered
the unbalanced sample size due to the different number of classes of EFTs (18) and PFTs (7) represented by FLUXNET2015
and the different number of EC sites per PFT class (which ranged between 3 and 31). To do this, we performed the following
steps:

First, we calculated all possible combinations (C) without repetitions between the 18 EFT and the 7 PFT classes (C(18,7) =
31834). Second, since the DA needs balanced data, we discarded all combinations with different numbers of EC sites in the
combined EFT and PFT classes. Third, for each combination, we applied discriminant analysis to assess how the EFT and PFT
classifications performed to discriminate the seasonal dynamics of NEE. For each discriminant analysis, we obtained five
metrics of performance (Table 2). Fourth, to assess whether significant differences existed in the performance metrics between
EFTs and PFTs, we applied the Wilcoxon non-parametric test. For each combination of a number of classes and EC sites, there
was a different number of discriminant analyses in the EFT subset and the PFT subset (S2 Table S1). To account for such an
unbalanced design during the Wilcoxon test, we fixed the sample size to the smaller subset (either from the EFT or the PFT
classification) and randomly bootstrapped the performance metrics from the bigger one. Fifth, we calculated the mean and
standard deviation of each metric obtained by the EFTs and PFTs classifications, the average p-value, and the percentage of
times we obtained significant differences (p-value <0.05) between EFTs and PFTs.

## 3 Results

### 3.1 Regional heterogeneity in ecosystem functioning using satellite-derived EFTs

The map of the EVI-derived proxies of productivity (EVI_mean), seasonality (EVI_SD), and phenology (DMAX) (S3 Fig.
S1a-c) and their integration into EFTs (Fig. 1) provided a characterization of the spatial patterns of our focal ecosystem function
across Europe. At the continental scale, productivity decreased eastwards and southwards (Fig. 1, S3 Fig. S4). Seasonality was



greater in cultivated and mountain grassland areas (Fig. 1, S3 Fig. S5), and the most frequent EVI maxima occurred in spring and summer (Fig. 1, S3 Fig. S6).

The greatest EVI_mean (D) was reached in the Atlantic and Continental biogeographic regions (Fig. 1, S3 Fig. S4d). At the same time, the lowest EVI_mean (A) occurred in the western part of the Mediterranean region, corresponding to most of the Iberian Peninsula, some parts of the Italian Peninsula, the mountainous areas of the Alpine region, and in the eastern part of the Continental region (Fig. 1, S3 Fig. S4a). The greatest seasonality (a) occurred in the highest altitudes of the Alpine region (peaks of Alps <3000 meters), the Continental region (southwestern, northwestern, and eastern part of this region), and the

eastern part of the Atlantic region (Fig. 1, S3 Fig. S5a). The lowest seasonality (d) was observed in the western part of the Mediterranean region, specifically in the Iberian Peninsula, the Gulf of Lion's surroundings, and the Atlantic region's Coastal western places (Fig. 1, S3 Fig. S5d). The phenological indicator of the growing season, DMAX, showed that most areas of the Mediterranean region have the EVI maxima in spring (1). EVI maxima in spring (1) were also observed in the Continental and Alpine regions (Fig. 1, S3 Fig. S6a). Maxima in summer (2) were identified in western places of the Atlantic and most of the

Alpine regions (Fig. 1, S3 Fig. S6b). EVI maxima in autumn (3) mainly in the Mediterranean region (Fig. 1, S3 Fig. S6c). Maxima in winter (4) were rare and emerged in the eastern part of the Atlantic region, where the maximum productivity was found and in the western part of the Mediterranean region (Fig. 1, S3 Fig. S6d).



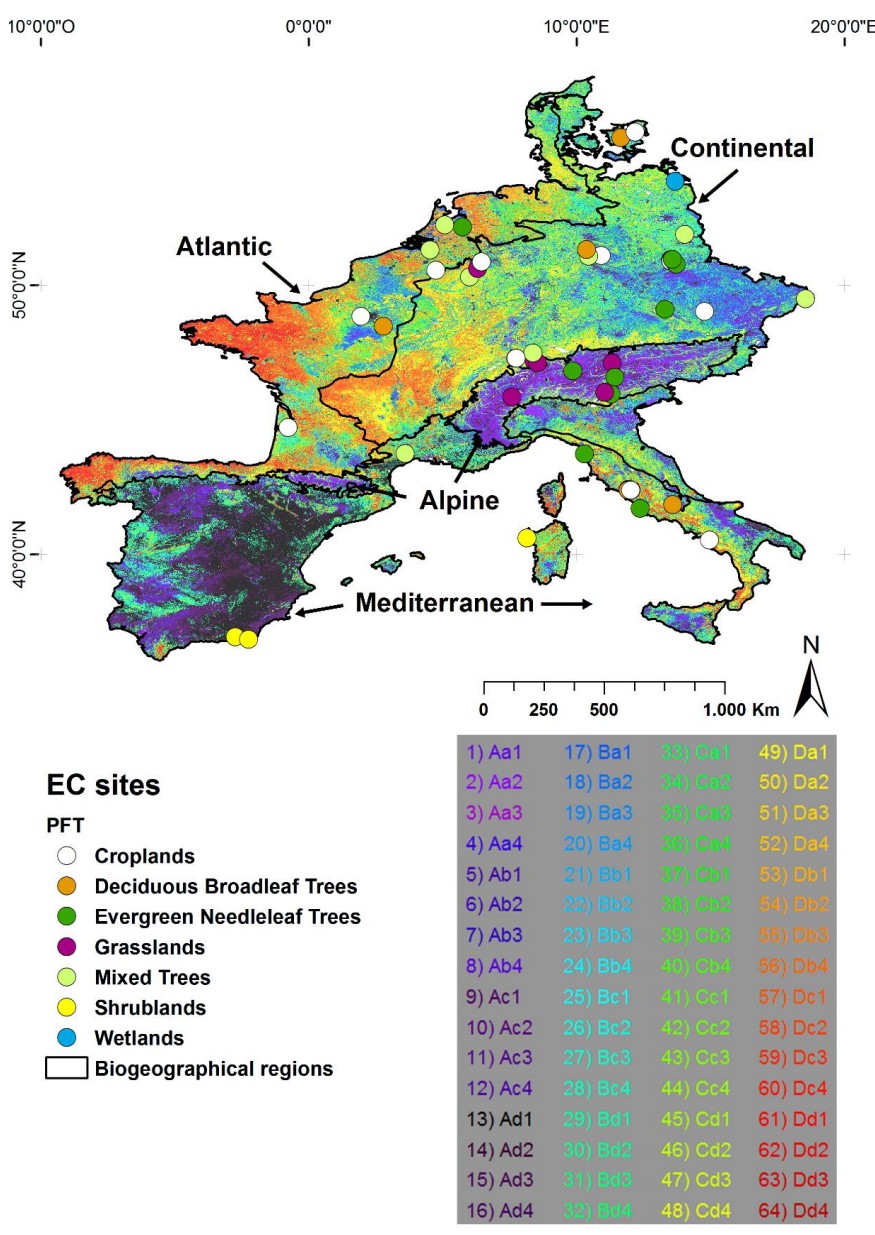

**Fig. 1.** Ecosystem Functional Types (EFTs) based on MODIS-EVI dynamics (~230 m resolution) and Eddy Covariance (EC) sites corresponding to the 2001–2014 period. Capital letters in the legend correspond to the EVI annual mean (EVI_mean) level, ranging from A to D for low to high productivity. Small letters show the seasonal standard deviation (EVI_SD), ranging from a to d for high to low seasonality of carbon gains. The numbers indicate the season when the maximum EVI took place (DMAX): (1) spring, (2) summer, (3) autumn, (4) winter. Places with EC sites are shown with squared colors, where each one represents a different plant functional type. Biogeographical regions are based on the official European biogeographical regions map (EEA, 2016) and are represented by black lines.





## 3.2 Ground-based NEE of the satellite-derived EFTs

In total, 20 of the 64 potential EFTs, containing 73.10 % of our study area, were represented by the network of the 50 long-term EC sites that met our selection criteria (Fig. 2). The most abundant EFT, Da1, showed high productivity (D), high seasonality (a), and maximum EVI in spring (1) (Fig. 2). Da1 occupied 10.87% of the surface and was distributed throughout

the study area but abundantly in the western and southern extremes of the Atlantic Region). Da1 was represented by 8 EC sites that exhibited NEE with a strong seasonal variability, with a pronounced peak of carbon assimilation between -7.23 and -7.46 g C m-2 d-1 in spring (Fig. 4) and corresponded with the most abundant ecosystem in Europe, the Deciduous Broadleaf and Mixed Forest (S2 Table S2). The second most abundant EFT, Ad1, showed low productivity (A), low seasonality (d), and maximum EVI also in spring (1). Ad1 occupied 9.98% of the territory, mainly in the Iberian Peninsula (Fig. 1). Ad1 was

represented by 2 EC sites (Fig. 2) that exhibited NEE dynamics with low seasonality and the peak of carbon assimilation (NEE) between -0.72 and -1.98 g C m-2 d-1 in spring (Fig. 4) and was concentrated in areas dominated by shrub vegetation (S2 Table S2).

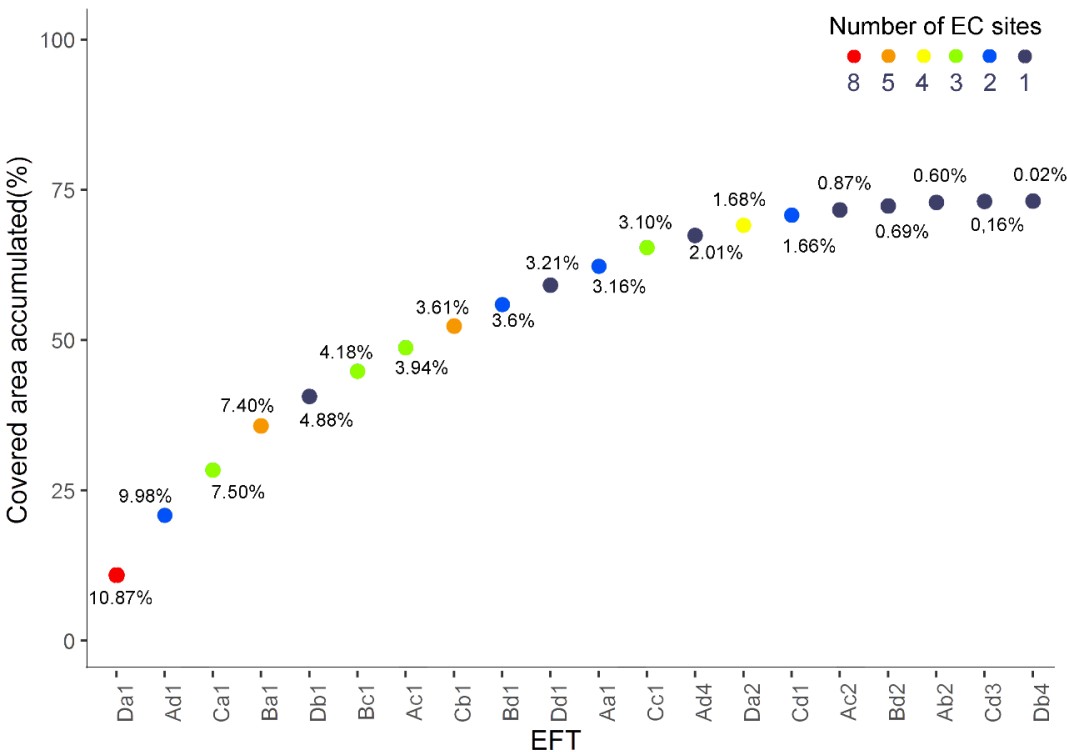

**Fig. 2.** Accumulated area covered by the Ecosystem functional types (EFTs; in %) represented in the study (ordered from highest to lowest). Colors indicate the number of eddy covariance (EC) sites, and the numbers indicate the area occupied by each of these EC sites (in %).





Regarding the abundance of EC sites, the EFT Da1 mentioned above was represented by 8 EC sites, followed by EFT Ba1 and
Cb1 with 5 EC sites. The EFT Ba1, was also abundant, occupying 7.4% of the total surface (Fig. 2), and was located mainly
in the eastern part of the study area (Atlantic and Continental regions) (Fig. 1). The EFT Cb1, was less abundant than the
previous one (3.61%) and was located in central areas of the Atlantic and Continental regions. NEE dynamics were
characterized by high (a) and medium-high (b) seasonality and the peak time of carbon assimilation between -6.40 and -7.53
g C m-2 d-1 in spring. In both cases, these places corresponded with cereal crops (S2 Table S2).

Our discriminant analysis showed that EFTs significantly differed in NEE measured in situ with the EC technique. The average
of the performance metrics obtained from the discrimination that satellite EFTs made of EC site NEE ranged between 0.953
to 0.978 (Table 3a). NEE dynamics significantly differed between different EFTs but were similar within the same EFTs (S3
Fig. S2). For example, the EFT "Da1", which had high productivity, high seasonality, and spring EVI maxima, also showed
high average NEE values, high seasonality in NEE, and maximum carbon assimilation in spring (Fig. 4, EC sites DE-Lnf, FR-
Fon). The EFT "Bc1", with medium to high productivity, medium seasonality, and spring EVI maxima, was also characterized
by moderate seasonality in terms of NEE and maximum carbon assimilation in spring (Fig. 4a for EC sites BE-Vie, DE-Tha).
Contrary, the EFT "Ad1", which had low productivity, low seasonality, and EVI spring maxima, also showed low average
NEE, low seasonality in NEE, and a peak of maximum carbon assimilation in spring (ES-Lju, IT-Noe). As another example,
the EFT "Cb1", with medium productivity, medium-high seasonality, and spring EVI maxima, also showed medium to high
seasonality in terms of NEE and maximum carbon assimilation in spring (Fig. 4a for EC sites DE-she, DE-RuS).

### 3.3 Comparison between EFTs and PFTs to discriminate NEE measured by EC

EFTs performed marginally better than PFTs in capturing differences in NEE dynamics measured on the ground (Table 3).
The average across all discriminant analyses in all performance indices was marginally but not significantly higher for EFTs
(e.g., mean Kappa = 0.953) than for PFTs (e.g., mean Kappa = 0.923) (Table 3, Fig. 3); However, the standard deviation (s.d.)
across all discriminant analyses was higher for PFTs (e.g., s. d. of Kappa = 0.078) than for EFTs (e.g., s. d. of Kappa = 0.067).
No significant differences between the performance metrics of EFTs and PFTs were detected by the Wilcoxon-test in any of
the indices (Table 3).





**Table 3.** Mean performances metrics, their standard deviation (SD) and differences in: Kappa, Accuracy, Precision, Recall and F1 values obtained from discriminant analysis of combinations with equal number of classes and EC sites of (a) ecosystem functional types (EFTs) and (b) plant functional types (PFTs). To assess for significant differences, we applied a Wilcoxon-test (p-values showed), and we calculated the percentage of cases in which differences between EFTs or PFTs with NEE were significant (% sig), in this case, none.

|  | a. EFTs | | b. PFTs | | Difference | |
| --- | --- | --- | --- | --- | --- | --- |
|  | mean | SD | mean | SD | p-value | % sig |
| **Kappa** | 0.953 | 0.067 | 0.923 | 0.078 | 1 | 0 |
| **Accuracy** | 0.972 | 0.040 | 0.952 | 0.051 | 1 | 0 |
| **Precision** | 0.967 | 0.047 | 0.959 | 0.057 | 1 | 0 |
| **Recall** | 0.978 | 0.033 | 0.960 | 0.040 | 1 | 0 |
| **F1** | 0.972 | 0.040 | 0.959 | 0.048 | 1 | 0 |



**Fig. 3.** Histograms of performances from discriminant analysis for all combinations of Ecosystem Functional Types (EFTs) and Plant Functional Types (PFTs) with equal number of classes and EC sites. Blue lines correspond to EFTs and green lines to PFTs.





In general, NEE dynamics were similar for the same PFT or EFT across EC sites (Fig. 4), though there were some exceptions for certain PFTs (Fig. 4b; S3 Fig. S3). Sites corresponding to the PFT "deciduous broadleaf trees" or the EFT "Da1" always showed similar NEE (Fig. 4; Table 1). However, for the PFT "evergreen needleleaf trees", NEE dynamics exhibited a different

seasonality and variable maximum carbon assimilation across sites (Fig. 4b for EC sites CH-Dav, DE-Lkb). Differences in NEE dynamics across sites were also observed for shrublands where the ES-LJu site (EFT Ad1) was assimilating carbon throughout the year, particularly in spring, while the ES-Amo site (EFT Ad4) was mainly emitting carbon throughout the year except for winter. Larger differences in NEE occurred in the PFT croplands, with maximum carbon sequestration occurring in different seasons (Fig. 4b, for sites CH-Oe1 and CH-Oe2 (EFT Cb1).


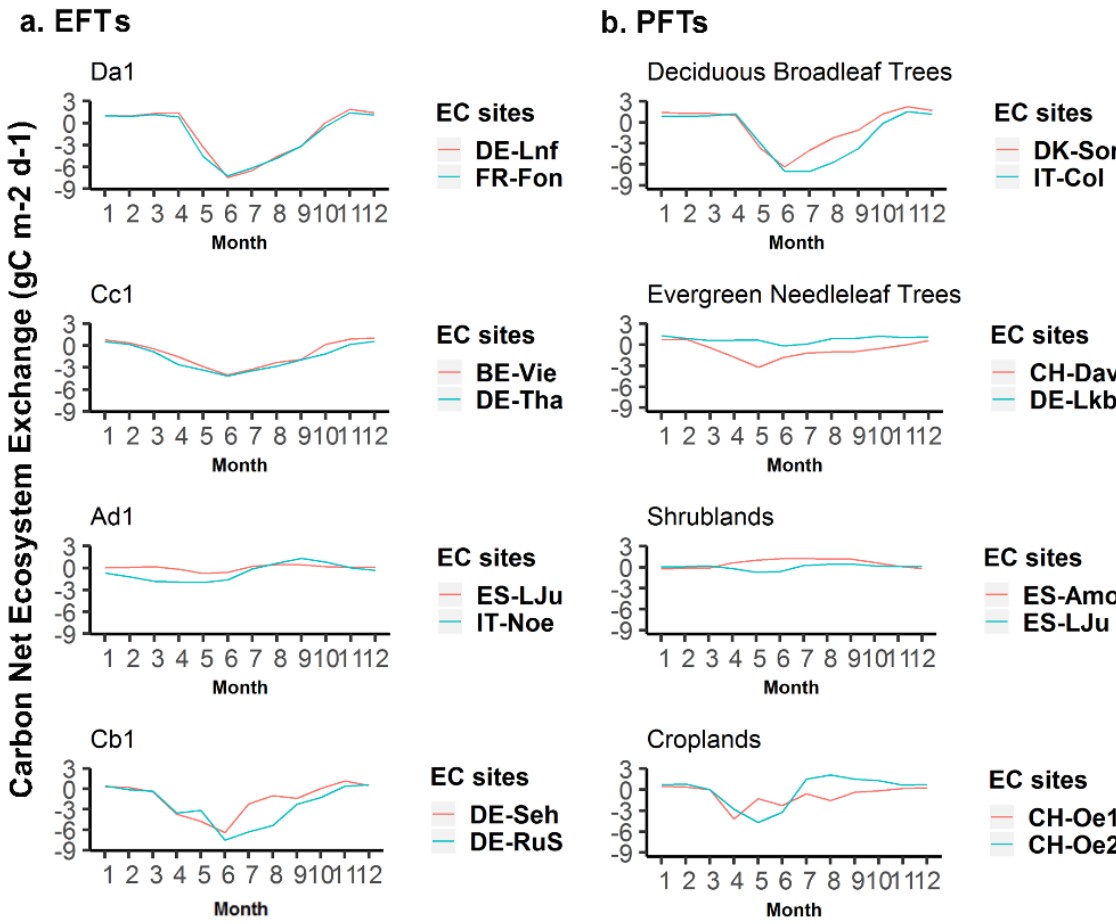

**Fig. 4.** Comparison of the variability within and across classes of Ecosystem Functional Types (EFTs) and Plant Functional Types (PFTs) in the seasonal dynamics of NEE. a) Variability inter EFTs: annual mean of NEE dynamics from different places



randomly selected with the same EFT; and b) variability inter PFTs and intra EFTs: annual mean of NEE dynamics from
different places with the same PFT and different EFT.

## 4 Discussion

Remotely-sensed EFTs successfully mapped functionally homogeneous land patches regarding NEE dynamics measured in
situ with the EC technique. Furthermore, EFTs performed at least similarly to the commonly used PFTs for discriminating
among different NEE seasonal dynamics (Table 3). EFTs have the advantage of being more sensitive in their responses to
short-term changes in ecosystem functioning than the slower-responding plant community composition or canopy structure.
Furthermore, they and can be recalculated on an annual basis using the same classification rules, which provides a
straightforward way to track interannual changes in ecosystem functioning (Müller et al. 2014). Our focal ecosystem function
was NEE dynamics, which is related to primary production (but also to ecosystem respiration), one of the most essential and
integrative descriptors of ecosystem functioning (Virginia and Wall, 2010). Hence, satellite-derived EFT classifications could
be used to monitor the status and changes of the regional heterogeneity or spatial diversity of the essential variable of ecosystem
productivity as a surrogate of the overall ecosystem performance (Jax, 2010; Pettorelli et al., 2016).

### 4.1 EFTs capture differences in NEE

EFTs quantified and mapped the spatio-temporal characteristics of carbon dynamics, a crucial aspect for biodiversity
conservation and ecosystem services maintenance in a global change context (Midgley et al. 2010). Twenty of the 64 EFTs
identified in Europe (corresponding to 73% of the study area) were represented by at least one EC site in the FLUXNET2015
dataset with at least three years of data. This number of site-years and the covered area provided sufficient evidence to confirm
the validity of the EFT concept. Therefore, our approach could help to assess carbon dynamics at a regional scale by providing
homogeneous land areas in terms of their primary production dynamics (Running et al. 2004, Zhang et al. 2015). This fact
helps to understand the regional patterns and drivers of the differences in carbon dynamics at the regional scale and could
contribute to reducing the uncertainties in the global carbon balance (Beer et al. 2010).

EFTs capture spatial differences in NEE seasonal dynamics equally well or marginally better than other mainstream
approaches, such as PFTs. Different areas may respond differently to environmental changes despite being dominated by the
same PFT, and frequently, ecosystem-process models (parameterized for a specific PFT) may not be able to represent these
differential responses (Vargas et al. 2013). Usually, the parameterization of a particular PFT is homogeneous within such PFT
and does not change, for instance, according to the eco-physiological status of a specific area or its intrinsic plasticity (Müller
et al. 2014). In addition, land-cover maps based on a PFT concept are static and difficult to update (i.e., PFT database structure
and assumptions are not easily adapted to new data). At the same time, EFTs are a data-driven classification through which
we can annually obtain new data and detect changes in the exchange of matter and energy between the ecosystems and the
atmosphere in response to environmental variability. In this sense, the literature (Bret-Harte et al. 2008; Suding et al. 2008;





Clark et al. 2016; Saccone and Virtanen 2017; Thomas et al. 2018) has pointed out that the PFT approach is not straightforward enough to represent ecosystem functional properties at the ecosystem level.

## 4.2 EFT spatial patterns and environmental controls

EFTs allowed us to characterize the regional heterogeneity of ecosystem functioning across Europe. In relation to the three descriptive attributes of ecosystem functioning from which the EFTs were constructed (EVI_mean; an estimator of primary

production, EVI_SD; a descriptor of seasonality and EVI_DMAX; an indicator of phenology), we found general patterns determined by the combination of vegetation characteristics and environmental controls. The role of environmental variables (abiotic and biotic) that control ecosystem processes differ according to the level of biological organization and the spatial scale considered (Reed et al. 1993; Pearson and Dawson, 2003). Ecosystem functioning in natural areas are known to be mainly driven by precipitation (Lauenroth et al. 1978), temperature (Rosenzweig and Dickinson 1968; Jobbagy et al. 2002), soil

characteristics (NoyMeir 1973), and vegetation structure (Epstein et al. 1998). In this case, EFTs productivity decreased from east to west influenced by rainfall patterns determined by the Gulf Stream and the distance from the ocean (Palter 2015), which also determines changes in vegetation. Regarding the seasonality of EVI, it increased in relation to two factors: 1) the altitude, having the highest values of seasonality in the mountainous areas (influenced by changes in precipitation, temperature, and consequently, in vegetation), and; 2) the crop areas, where management practices, harvests, and crop changes are responsible

of this dynamic and therefore it cannot be explained by natural environmental controls alone. Peaks of maximum EVI in Europe took place in spring and summer when the availability of water (precipitation) and energy (temperature) for vegetation was at its optimum (Whittaker et al. 2003).

Boundaries of the biogeographical regions (EEA 2016) were consistent with the EFTs (Fig. 1). Still, while the classification from EEA is static, EFTs provide a data-driven classification that could be better coupled to ecosystem functioning. The Alpine

region was dominated by EFTs with low productivity, high seasonality, and maxima in summer. In the high mountain peaks (<3000 meters), the vegetation was reduced to a low density of highly adapted plants that can tolerate extreme conditions (i.e., the short growing period and fluctuating air temperatures, and therefore, has low productivity, also detected in the global primary productivity patterns of Beer et al. (2010) and Zhang et al. (2017)). In the highest altitudes, snow is present over most of the year, leaving only a short period for the development of the plants, mainly in summer, leading to a summer maximum

and a high seasonality (Sundseth, 2009a).

A high heterogeneity of EFTs characterized the Mediterranean region due to their high habitat diversity (i.e., high mountains and rocky shores, thick scrub and semi-arid steppes, coastal wetlands, and sandy beaches, constituting a global biodiversity hotspot (Myers et al. 2000)). The main driver of ecosystem functional diversity is the climate (characterized by hot, dry summers and cool winters) (Lionello et al. 2006), in combination with human influence, (i.e., livestock grazing, forest

cultivation, and forest fires) (Blondel and Aronson, 1999).

The Atlantic region was characterized by EFTs with high productivity, high seasonality, and maximum greening in spring due to the mild winters, cool summers, predominantly westerly winds, and moderate rainfall throughout the year (Hurrel, 1995).



These conditions favor non-water-limited deciduous species with high productivity, resulting in a high seasonality. Due to the anthropogenic influence, agricultural landscapes are widespread in this region, one of Europe's five major agricultural regions,

according to Kostrowicki (1991). Thus, the region's high productivity must be partly attributed to irrigation, and high seasonality is driven by harvest and cropping cycles.

Finally, in the Continental region, the ecosystem's functioning varied largely in terms of productivity, reflecting regional climatic patterns. In the eastern part of the continental region, extremes of hot and cold temperatures and wet and dry conditions are more frequent and strongly impact ecosystem functioning (dominant EFT was Aa1, low productivity, high seasonality, and

maximum in spring). These areas are mountainous and experience sub-alpine conditions. Moving west, the climate is characterized by relatively small temperature fluctuations due to the buffering effect of the nearby ocean and the flat landscape (Da1 and Ca1 in the transition) (Sundseth, 2009b).

## 4.3 Opportunities and limitations of EFTs

Since EFTs describe ecosystem functioning on an annual basis in homogeneous patches on the land surface, they offer

opportunities for application in ecology and conservation compared to approaches that do not represent short-term dynamics (such as PFTs). However, they also have some limitations.

The concept of EFT has been highlighted as "the first serious attempt to group ecosystems (at large scales) based on shared functional behavior" (Mucina, 2019), and its strength for being applied as a classification scheme is determined by its ability to translate ecosystem functions into discrete entities that can be mapped. EFTs are identified by remote sensing tools from

aggregated measurements of ecosystem functions at the pixel level, which, in practice, represents information on the performance of the whole ecosystem at that grain scale. Having the possibility of mapping entities (EFTs) that reflect the principal performance of the entire ecosystem opens a straightforward, tangible, and biologically meaningful way to quantify distributions of ecosystem functions at the regional scale, complementing our traditional view of ecosystems (Paruelo et al. 2001; Butchart et al. 2010; Asner et al. 2017). Specifically, satellite-derived dynamic functional classifications, such as EFTs,

have several advantages over other static approaches, such as PFTs. Satellite-derived EFAs and EFTs 1) are capable of capturing differences in ecosystem processes as measured in the field; 2) they provide a valuable framework for understanding the mechanisms underlying large-scale ecological changes (Cabello et al. 2016; Alcaraz-Segura et al. 2017; Requena-Mullor et al. 2017, 2018; Arenas-Castro et al. 2018; Lourenço et al. 2018; Vaz et al. 2018); 3) they offer a faster response than compositional or structural approaches to environmental changes (McNaughton, 1989; Mouillot et al. 2013), which are

particularly noticeable at the ecosystem level (Vitousek, 1994); 4) they can be more easily monitored and updated than structural or compositional ones under a common protocol in space and time, at different spatial scales and over large extents (Paruelo et al. 2001); 5) they can complement information on vegetation structure and composition (e.g., canopy architecture, vegetation type, PFT), because they constitute complementary dimensions of biodiversity complexity (Noss, 1990); 6) they facilitate the direct assessment of ecosystem functions and services (Costanza et al. 2006; Hellmann et al. 2017)

and would link critical dimensions of biodiversity to ecosystem processes including the carbon cycle, the water cycle and the



provisioning of ecosystem services; 7) they have already been proposed as essential variables for monitoring biodiversity (Pettorelli et al. 2016; Skidmore et al. 2021).

Our approach, as with any other ecosystem classification framework, is still subject to some challenges. First, EFTs represented by several EC sites could be parameterized in terms of NEE dynamics, though not all EFTs (18%) are represented yet. Second, the footprint or spatial resolution of the EC measurements varies depending on the micrometeorological conditions (wind direction, wind speed, atmospheric stability) and the ratio of measurement to vegetation height, e.g., forest flux footprints are generally larger than grassland footprints (oscillates between 50 m and 200 m). In comparison, the MODIS pixels used have a constant spatial resolution of ~231 m. Such limitations could be handled in future works using satellites with higher spatial resolution, such Sentinel-2 (10 m/pixel), but currently is not possible because the time period of Sentinel-2 data is not covered by FLUXNET data (i.e., Sentinel-2 starts taking data in 2015 and the available FLUXNET database goes up to this year). Third, different ecosystems regarding other functional aspects (e.g., evapotranspiration, heat exchange) can be classified here as the same EFT from the NEE dynamics, as we used it as our focal function. However, EFTs could also be identified to characterize the spatiotemporal heterogeneity of multiple ecosystem processes and functions at different scales, including other functional aspects (e.g., albedo, evapotranspiration, heat exchange) (Fernandez et al. 2010). Finally, incorporating EFTs into earth system models is challenging since these models can use simple and small numbers of categories in a variable, and some models might not be able to run with so many (64) EFT categories. Nevertheless, some studies have successfully incorporated EFTs into earth system models (Lee et al. 2013; Müller et al. 2014). The incorporation of these types of variables (dynamic and easily accessible) into the models might be helpful in the monitoring and sustainable management of carbon reservoirs at short to medium-time scales.

**5 Conclusion**

Satellite-derived EFTs are an ecosystem functional classification built from satellite observations of radiation exchanges between the land surface and the atmosphere that allow the identification of homogeneous land patches in terms of an essential ecosystem function, e.g., NEE dynamics, measured on the ground by means of which is related to ecosystem productivity. EFTs performed as well as PFTs in discriminating different NEE dynamics, EFTs, however, have two main advantages: they can be easily updated for any region of the world at an annual frequency based on available satellite information, and EFTs maps are more sensitive to environmental changes than vegetation composition or structure.

Our results showed the capability of using ecosystem functional attributes for grouping ecosystems at large scales according to their different net carbon flux dynamics. Such classification, based on the essential biodiversity variable of ecosystem production as a focal ecosystem function, opens the possibility of assessing and monitoring ecosystem functional diversity, the spatial heterogeneity in ecosystem functioning, and carbon-related ecosystem services at regional to global scales. Therefore, our study demonstrates that satellite-derived EFTs provide a valid tool to assess and monitor ecosystem functioning with potential applications in ecosystem monitoring and modeling and biodiversity and carbon management programs.




**Data availability**

The MODIS database used in this work is maintained by NASA (satellite Terra, sensor MODIS, product MOD13Q1.006) and is mirrored by Google on the Earth Engine servers ([https://developers.google.com/earth-engine/datasets/catalog/MODIS_006_MOD13Q1](https://developers.google.com/earth-engine/datasets/catalog/MODIS_006_MOD13Q1)). FLUXNET2015 eddy covariance data are available through the FLUXNET([https://fluxnet.org/data/fluxnet2015-dataset](https://fluxnet.org/data/fluxnet2015-dataset)). The Google Earth Engine code used to derive Ecosystem Functional Types (EFTs) is openly available at [https://doi.org/10.5281/zenodo.7524973](https://doi.org/10.5281/zenodo.7524973). The plant functional types (PFTs) used in this

study are based on the IGBP-DIS global 1 km land cover data set "DISCover": proposal and implementation plans, IGBP-DIS available at [https://daac.ornl.gov/ISLSCP_II/guides/edc_landcover_xdeg.html](https://daac.ornl.gov/ISLSCP_II/guides/edc_landcover_xdeg.html).

**Author contributions**

DAS, AM, JC, JP and BPC designed the study, AM and DAS coordinated it. BPC processed the data and prepared the
manuscript with contributions from all authors. BPC and JML prepared the final Fig.s. LM, AK, LS, BG, JD, LŠ, AI, GW, EP, KF, AM, MP, LM, LH, PD, IG, and KP provided FLUXNET data. All authors reviewed the article and provided valuable feedback, especially RV and JML.

**Competing interests.** Some authors are members of the editorial board of journal Biogeosciences.

**Acknowledgements.** This publication is part of the EVEREST project (PID2023-151939OB-I00) funded by MICIU/AEI/10.13039/501100011033 and by ERDF/EU. Funds were also provided by ERDF and Spanish MINECO (project CGL2014-61610-EXP) and to B.C. by University of Almería (PhD contract: research training program). ECOPOTENTIAL, which received funding from the European Union's Horizon 2020 Research and Innovation Program under grant agreement No. 641762, and the NASA 2016 GEOBON Work Programme Grant # 80NSSC18K0446. EarthCul (reference PID2020-
118041GB-I00), funded by the Spanish Ministry of Science and Innovation, Smart-EcoMountains, LifeWatch-ERIC action line, within the Workpackages LifeWatch-2019-10-UGR-01_WP-8, LifeWatch-2019-10-UGR-01_WP-7, and LifeWatch-2019-10-UGR-01_WP-4; This work used eddy covariance data acquired and shared by the FLUXNET community. The ERA-Interim reanalysis data are provided by ECMWF and processed by LSCE. The FLUXNET eddy covariance data processing and harmonization was carried out by the European Fluxes Database Cluster, AmeriFlux Management Project, and Fluxdata
project of FLUXNET, with the support of CDIAC and ICOS Ecosystem Thematic Center, and the OzFlux, ChinaFlux and AsiaFlux offices. JML was funded by the Plan Propio de Investigación (P9) of the University of Granada. AM was funded by the Deutsche Forschungsgemeinschaft (DFG, German Research Foundation) under Germany's Excellence Strategy – EXC 2070 – 390732324. LM acknowledges the funding provided by Forest Services, Autonomous Province of Bolzano. LŠ acknowledges support from the Ministry of Education, Youth and Sports of the Czech Republic within the CzeCOS program
(grant number LM2023048) and the AdAgriF project (CZ.02.01.01/00/22 008/0004635).



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
