# Peer review of "Satellite-derived Ecosystem Functional Types capture ecosystem functional heterogeneity at regional scale"

_EGUsphere, 2025_

## Referee Comment (RC2)

**General assessment:**

This study provides a valuable perspective on how to evaluate diversity in ecosystem functional patterns through the classification of Ecosystem Functional Types (EFTs). EFTs are surface-based classifications derived from key functional attributes of ecosystems, and here they are compared against more conventional classifications that emphasize composition and structure, such as Plant Functional Types (PFTs). The comparison is grounded on in situ measurements obtained with the eddy covariance technique, one of the most robust and reliable approaches for quantifying ecosystem-level functional processes. The analysis further benefits from the use of the highly reputable FLUXNET2015 database.

The methodological design is rigorous, offering a well-balanced comparison between EFTs and PFTs prior to subjecting them to discriminant analysis. Although results did not reveal statistically significant differences, the study successfully validates the effectiveness of EFTs in representing functional patterns at the ecosystem scale. A key advantage is that EFTs provide more dynamic insights than classifications based solely on compositional or structural traits.

The authors also acknowledge the limitations of EFTs, particularly regarding spatial resolution and the fact that not all EFT classes were represented in the study area. These issues are appropriately addressed in Section 4.3. Furthermore, the manuscript is well-written, demonstrating good coherence, cohesion, and flow.

**Concluding remarks:**

This work makes an important contribution to the representation of spatial functional patterns and their comparison against more conventional classification systems, validated with in situ flux measurements. The choice of the study domain is appropriate given the high density of flux tower sites. However, due to the diversity of EFTs, not all classes were represented by field observations. Nevertheless, more than 70% of the spatial coverage of EFTs was captured within the study area. Overall, I consider this manuscript to be in a publishable form as it stands.

---

## Author Comment (AC1)

**Response to Reviewer Comments**
**Manuscript title: Satellite-derived Ecosystem Functional Types capture ecosystem functional heterogeneity at regional scale**

**Authors: Beatriz P. Cazorla et al.**

**General assessment:**

This study presents a timely analysis of ecosystem functional heterogeneity across Europe by testing whether satellite-derived Ecosystem Functional Types (EFTs), defined from MODIS EVI time series, are coupled with Net Ecosystem Exchange (NEE) patterns measured at eddy-covariance (EC) sites. By comparing EFTs with conventional Plant Functional Types (PFTs), the authors explore whether remote-sensing-based classifications provide a more dynamic alternative for ecosystem functional monitoring. The paper is well-structured, clearly written, and addresses an important research gap in functional biogeography. The authors provide robust empirical evidence across 50 EC sites and multiple biogeographic zones, reinforcing the potential of EFTs to serve as integrative descriptors of carbon dynamics. The methodological framework is rigorous, and the discussion is thoughtful and comprehensive. However, several methodological choices and interpretative aspects could benefit from clarification, expansion, or further justification.

……………………………………………………………………………………………………

RESPONSE TO COMMENTS

……………………………………………………………………………………………………

Dear reviewer,

We thank the reviewer for the positive and encouraging evaluation of our work, as well as for the constructive comments that will provide valuable insights to further strengthen the conceptual and methodological robustness of our manuscript. In our response below, please find our point-by-point responses (indicated with "R") presenting, in detail, how we have addressed the Reviewer's comments ("C").
* * *
**Major comments**

**C1 - *The study focuses exclusively on EVI-derived EFTs as proxies for ecosystem functioning, which primarily capture carbon uptake via vegetation greenness. This focus, while justified, represents only one dimension of ecosystem function. Consider acknowledging more explicitly in the introduction and discussion that EFTs in this implementation reflect carbon-related dynamics. The authors should also consider whether incorporating additional functional attributes (e.g., NDWI for water stress, land surface temperature, albedo, and evapotranspiration) could enhance EFT robustness, particularly in water-limited ecosystems such as the Mediterranean region.**

R1 - We thank the reviewer for the suggestions, which will be implemented in the new version of the manuscript. Below we detail why we focus on EVI-derived EFTs as proxies for ecosystem functioning.

First, we chose EVI instead of any other vegetation index as a surrogate for carbon gain dynamics since EVI is assumed to be more reliable in both low and high vegetation cover situations (Huete et al., 1997). EVI is sensitive to changes in areas having high biomass, EVI reduces the influence of atmospheric conditions on vegetation index values, and EVI corrects for canopy background signals. At present, several models use EVI or NDVI to estimate APAR (Absorbed Photosynthetically Active Radiation) and NPP (Net Primary Production), such as CASA (Potter et al., 2007) or MOD17 (Running and Zhao. 2015).

Second, remote sensing-based models that estimate NPP or GPP from vegetation indices are based on a PFT-driven land-cover map legend. This implies that the potential errors in the land-cover map and in the models will propagate to the NPP and GPP products (as shown by Zhu and others 2016; Wang and others 2017). Several studies suggest that EVI provides a straightforward but robust approach to estimating spatial patterns of global annual GPP (e.g.,Shi and others 2017).

Third, using EVI instead of MODIS GPP or NPP has the additional advantage of offering a finer spatial resolution: 230 m for EVI compared to 500 m for NPP. The only NPP product with a resolution close to 250 m is available exclusively for the USA.

Finally, we decided to start with vegetation indices as surrogates for primary production dynamics since primary productivity represents the energy that enters into the life cycle, it is the most integrative indicator of ecosystem functioning (Virginia and Wall, 2001), and it is linked to multiple ecosystem processes and services (Paruelo and others 2016). In future studies, we will incorporate other dimensions of ecosystem functioning apart from the carbon gains dynamics, such as the sensible heat dynamics (e.g., surface temperature), latent heat dynamics (e.g., evapotranspiration), radiative balance (e.g., albedo), water dynamics (e.g., evapotranspiration), disturbance regime (e.g., flood and fire dynamics), etc.

Following the reviewer's suggestion, we will add to the introduction and the discussion that EFTs in this implementation reflect carbon-related dynamics and we will also mention the potential of incorporating additional functional attributes such as NDWI, land surface temperature, albedo, and evapotranspiration in future work to enhance EFT robustness.

**C2 - \* The current approach partitions EVI_mean, EVI_SD, and EVI_DMAX into four bins each, generating 64 EFT classes. However, the justification for choosing four intervals remains vague, and it is unclear how sensitive the results are to this choice. Could you please clarify the rationale behind selecting four intervals per metric? Would the patterns hold if three or five bins were used instead? A supporting table defining the intervals or example ranges for each bin would significantly improve interpretability.**

R2 - The selection of four intervals per metric is based on ecological interpretability and methodological precedent. For EVI_DMAX, four intervals correspond directly to the four seasons of the year. For EVI_mean and EVI_SD, we calculated annual quartiles averaged over the study period, ensuring balanced class representation and comparability with previous EFT studies (e.g., Alcaraz-Segura et al., 2013, Cazorla et al. 2020, 2023).

Following the functional classification principles of Noble and Gitay (1996), we began with the simplest structure that remained ecologically interpretable. Using four categories per variable

produced 64 potential EFT classes (4 × 4 × 4), which is manageable, preserves observed spatial patterns, and allows straightforward interpretation of the legend. In theory, finer divisions (e.g., 23 × 23 × 23 = 12,167 potential classes) are possible but would be impractical for analysis and interpretation.

We will expand the rationale in Section 2.2 and add a new table in the Supplementary Material showing representative value ranges for each interval.

**C3 - *While visually appealing, the EFT map (Fig. 1) is difficult to interpret due to the high number of classes. The dense legend makes it hard to discern regional patterns or relate the map to key findings. Consider providing a simplified version of the map by aggregating the EFTs into broader clusters (e.g., via PCA, hierarchical clustering, or functional similarity groupings).**

R3 - We appreciate this suggestion. In the Supplementary Material, we could add a simplified version of the EFT map created by clustering EFTs. This version will facilitate the visual interpretation of large-scale patterns, while the full-resolution classification is retained in the main text for methodological completeness.

**C4 - *The authors analyse NEE seasonal dynamics as the basis for comparing EFTs and PFTs. However, ecosystem function varies across multiple temporal scales. Please clarify why only seasonal cycles were analysed. Could complementary metrics, such as daily anomalies, interannual variability, or cumulative annual fluxes, provide additional insight into functional distinctiveness across EFTs?**

R4 - We chose to focus on seasonal cycles because they are directly linked to our EVI-based EFT framework and provide a robust temporal signal given the resolution and length of MODIS and FLUXNET data series. Nonetheless, we agree that other temporal metrics (e.g., daily anomalies or interannual variability) could provide additional insights. We will add this point to the Discussion as a potential extension of the study.

**C5 - *The MODIS spatial resolution (~230 m) does not always match the EC tower footprint (~50–200 m), which varies depending on meteorological conditions and site characteristics. Please address whether a footprint-weighted EVI averaging was considered or feasible. At a minimum, discussing the potential impact of footprint mismatch on EFT-NEE comparisons would enhance methodological transparency.**

R5 - We acknowledge this limitation. Footprint-weighted averaging was not applied because daily footprint data were not available for all sites and years in our study. We will add a paragraph in the Discussion explicitly addressing the potential impacts of footprint mismatch on EFT–NEE comparisons.

MODIS was selected over higher-resolution sensors such as Landsat or Sentinel-2 for several reasons. Sentinel-2 data are only available from 2015 onwards, whereas the FLUXNET dataset we used ends in 2015, preventing temporal overlap. Landsat offers higher spatial resolution but acquires only one

image every 15 days; in the European climate, this low temporal frequency combined with frequent cloud cover would result in a high proportion of missing or low-quality observations. MODIS, in contrast, provides a consistent, cloud-screened, 16-day composite time series over the full study period, ensuring data continuity and comparability.

We will note in the Discussion that future work could address scale mismatches by using higher-resolution sensors (e.g., Sentinel-2) or footprint modeling, provided sufficient temporal overlap and cloud-free data are available.

**C6 - \*With 64 possible EFTs, only 20 are represented in the EC network. This granularity may be problematic for integration into Earth system models, which typically rely on a smaller number of categories. Have the authors considered simplifying the EFT classification, for example, by grouping rare classes or employing dimension-reduction techniques? Providing a roadmap for EFT integration into models would enhance the study's relevance.**

R6 - We agree that the high granularity of the full EFT classification could pose challenges for direct integration into Earth system models. However, it is worth noting that the 20 EFTs represented in the EC network are also the most abundant in the study area, together covering 73.10% of the total surface. This means that our EC network captures the dominant functional types across Europe, ensuring that the most ecologically relevant classes are well represented in the analysis.

**C7 - \*The study uses EVI_DMAX as a phenology metric. However, the start and end of the growing season are also informative indicators of functional timing and duration. Please clarify whether metrics such as SOS/EOS (start/end of season) were tested or considered. If not, do the authors anticipate that they could provide complementary or better information than EVI_DMAX?**
R7 - We did not use SOS/EOS in this analysis because EVI_DMAX is less sensitive to interannual noise and is directly aligned with our hypothesis on the timing of peak productivity. However, we agree that SOS/EOS could provide complementary information on growing season duration and timing. We have included this consideration in the Discussion of the revised version of the manuscript as a future methodological extension.
* * *
**Minor comments**

**C8 - \*L66: Please clarify which method is referenced for estimating EFTs from EC measurements.**

R8 - We will clarify in the text the method used to derive EFTs from EC measurements.

**C9 - \* L138 144: The sentence describing the naming convention is dense and complex to digest. A schematic or table showing example combinations (e.g., Ba1, Cb2, etc.) and their meaning would be helpful for unfamiliar readers.**

R9 - We appreciate the reviewer's comment. Examples of the EFT codes together with their meaning are already included in the main text (lines 223 and 228). Nevertheless, to further improve clarity, we

will add a schematic table in the Supplementary Material summarizing several examples along with their corresponding meaning. We will also simplify this sentence.

**C10 - *Caption of Fig. 1: The caption refers to squared colours, but circles appear to be used instead. Please correct for consistency.**

R10 - We thank the reviewer for pointing out this inconsistency. The caption of Fig. 1 will be corrected by replacing "squared colours" with "circles" to ensure consistency with the figure.

**C11 - *Concluding remarks**

**This study makes a valuable contribution to the growing body of literature on remote sensing of ecosystem function. The empirical validation of EFTs against eddy-covariance NEE across Europe is a significant achievement, and the comparisons to PFTs are well-executed and relevant. The manuscript would benefit from more explicit justifications for methodological choices (e.g., the number of intervals), a more nuanced discussion of scale mismatches and model usability, and an exploration of additional dimensions of ecosystem functioning.**

R11 - We are grateful for the reviewer's constructive suggestions, which will strengthen our manuscript. The changes will include explicit methodological justifications, a clearer discussion of limitations (number of intervals, scale mismatches, model usability), the provision of additional visualizations, and the recognition of future research directions.

---

## Author Comment (AC2)

**RC2: 'Comment on egusphere-2025-2835', Samuel Villarreal, 07 Sep 2025   reply**

We thank the reviewer for the careful reading and the positive and constructive assessment of our manuscript. Below we provide a point-by-point reply, where reviewer comments are indicated with "C" and our responses with "R."

**C1 - \*General assessment: This study provides a valuable perspective on how to evaluate diversity in ecosystem functional patterns through the classification of Ecosystem Functional Types (EFTs). EFTs are surface-based classifications derived from key functional attributes of ecosystems, and here they are compared against more conventional classifications that emphasize composition and structure, such as Plant Functional Types (PFTs).**

R1 - We appreciate this recognition. Our objective was precisely to examine whether classifications based on EFTs could complement or match PFTs classifications, offering a dynamic characterization that could be updated annually and was feasible to measure at the global scale. We were pleased that this comparison was considered valuable.

**C2 - \*The comparison is grounded on in situ measurements obtained with the eddy covariance technique, one of the most robust and reliable approaches for quantifying ecosystem-level functional processes. The analysis further benefits from the use of the highly reputable FLUXNET2015 database.**

R2 - We thank the reviewer for highlighting this. These datasets are indeed essential for ensuring reliability, and we are glad that their inclusion strengthens confidence in the study.

**C3 - \*The methodological design is rigorous, offering a well-balanced comparison between EFTs and PFTs prior to subjecting them to discriminant analysis.**

R3 - We appreciate this comment. Our intention was to establish the correct framework for comparing EFTs and PFTs, and we are pleased that this was recognized.

**C4 -\*Although results did not reveal statistically significant differences, the study successfully validates the effectiveness of EFTs in representing functional patterns at the ecosystem scale. A key advantage is that EFTs provide more dynamic insights than classifications based solely on compositional or structural traits.**

R4 - We agree with this interpretation. The comparable performance of EFTs and PFTs shows that EFTs are able to capture functional patterns reliably. Furthermore, a key advantage is that EFTs provide more dynamic insights than classifications based solely on compositional or structural traits.

**C5 - \*The authors also acknowledge the limitations of EFTs, particularly regarding spatial resolution and the fact that not all EFT classes were represented in the study area. These issues are appropriately addressed in Section 4.3.**

R5 - We are glad that our discussion of these limitations was found appropriate. While some classes were missing, we agree that the coverage of >70% provides robust support for our conclusions.

**C6 - * Furthermore, the manuscript is well-written, demonstrating good coherence, cohesion, and flow.**

**R6 -** We are very grateful for this positive feedback on the clarity of the manuscript.

**C7 - *Concluding remarks:**
**This work makes an important contribution to the representation of spatial functional patterns and their comparison against more conventional classification systems, validated with in situ flux measurements. The choice of the study domain is appropriate given the high density of flux tower sites. However, due to the diversity of EFTs, not all classes were represented by field observations. Nevertheless, more than 70% of the spatial coverage of EFTs was captured within the study area.**

R7 - We thank the reviewer for acknowledging this. We are pleased that the coverage achieved is considered sufficient.

**C8 - *Overall, I consider this manuscript to be in a publishable form as it stands.**

R8 - We sincerely thank the reviewer for this positive conclusion.